# Failure Analysis for Gold Wire Bonding of Sensor Packaging Based on Experimental and Numerical Methods

**DOI:** 10.3390/mi14091695

**Published:** 2023-08-30

**Authors:** Yameng Sun, Kun Ma, Yifan Song, Tongtong Zi, Xun Liu, Zheng Feng, Yang Zhou, Sheng Liu

**Affiliations:** 1Laboratory for Electronic Manufacturing and Packaging Integration, The Institute of Technological Sciences, Wuhan University, Wuhan 430070, China; sunyameng@whu.edu.cn (Y.S.); mk_175@whu.edu.cn (K.M.); xun.liu@whu.edu.cn (X.L.); 2School of Mechanical Science and Engineering, Huazhong University of Science and Technology, Wuhan 430070, China; syfmvp@163.com; 3Hefei Archimedes Electronic Technology Co., Ltd., Hefei 230000, China; berry.zi@ac-semi.com (T.Z.); fzheng_kite@163.com (Z.F.); 4School of Power and Mechanical Engineering, Wuhan University, Wuhan 430070, China

**Keywords:** failure mechanism and reliability, wire bonding, wire pull and shear test

## Abstract

There is an increasing demand for the use of automotive sensors where complex working environments may easily lead to failure. Wire pull and shear test models based on finite-element analysis are established to evaluate their reliability by investigating the failure mode and mechanism of gold wire bonding. The effect of shear force position and pull force position on failure is also analyzed. The bonding failure was verified by experiments, which is consistent with the simulation result. The results show that: (1) The three-dimensional quantitative modeling reveals the process of bonding delamination and stress concentration. (2) The bonding–slip method (BSM) is adopted in the gold ball detaching process. The concept of three states, including deformation accumulation, cracking, and disengagement, was put forward to reveal the interface stress evolution trend according to the shear testing results. The results indicate that in the interface, the stress in the deformation accumulation state decreases from the tensile side (or compression side) to the center, and the stress in the cracking and disengagement states reduces gradually from the tensile side to the edge. When the interface is completely separated, the failed shear force concentrates on 42 g. The concept and theory proposed in this work can effectively reveal the failure mechanism of bonding interface and help to establish a new failure criterion.

## 1. Introduction

The reliability of automotive sensors can be evaluated and improved by predicting their health status. Complex environmental factors, such as ambient temperature, random vibration, drop, etc., have a great impact on the reliability and lifetime of electronic components, including sensors. In this work, a sensor reliability analysis model is established according to the certification requirements of the Automotive Electronics Committee (AEC-Q100). The model describes the wire bonding stress state of the pressure sensor MLX90807 under different reliability testing standards. Through the combination of numerical and experimental methods, the effectiveness of qualitative and quantitative analysis of automotive sensor failure evaluation can be improved.

With the rise of 5G communication and the internet of things, advanced driving assistant aystems (ADAS) and on-board entertainment are gradually becoming the standard configuration of new-generation intelligent vehicles. Automotive sensors are the input devices of automobile computer systems, which convert the various operation states of a car (such as vehicle speed, medium temperature, engine operating conditions) into electric signals and send them to a central processing unit (CPU), putting the engine in the optimal working conditions. Various automotive sensors are distributed throughout the operation systems, and therefore, much attention should be paid to their reliability [1]. Sensors can be divided into temperature, pressure, flow, position, gas concentration, speed, brightness, dry humidity, distance, etc. based on their different functions [2]. The pressure sensor takes advantage of three strain gauge technologies, a ceramic capacitor and micro-electromechanical and glass micro-melted silicon, to satisfy the requirements of different system environments [3]. Automotive sensors used in intelligent vehicles usually work in relatively harsh environments, such as random vibration, shock/high shock, and extreme temperature. Once the sensor fails, the respective device cannot perform well. Therefore, sensors play a vital role in automobiles.

The packaging and interconnection are the keys to affecting the reliability and performance of automotive sensors. According to the JESD22-A104D specifications, the mismatch of the coefficient of thermal expansion (CTE) will cause significant thermo-mechanical stress on the key bumps under the temperature load of packaging components [4,5,6,7,8,9,10]. Wire bonding is used to connect the input and output terminals of the chip with corresponding pins by using the good conductivity and ductility of metal wires to achieve circuit interconnection. Wire bonding technology, as the most basic interconnection technology, has a great influence on device reliability. Circuit breakage and even chip damage often occur if the bonding force is set improperly. The optimization rules obtained through the finite-element method (FEM) can greatly improve device working efficiency and save costs. Quite a lot of systematic research has been performed on the reliability of wire bonding. Some new and innovative ideas [11] are introduced to address the packaging processing reliability, which was elaborated fully and addressed timely and widely in the community. The methodology and results cover a good and solid approach combining experimental results, analytical modeling, and simulation modeling. Zhang and Lee [12] conducted a numerical analysis to evaluate the fatigue life of the gold wire bond in the package. Krause et al. [13] evaluated the stress behavior of pressure sensors under mechanical shock. Ren et al. [14] developed a simulation model to predict the temperature distribution and the associated thermal stresses of an amperometric oxygen sensor during the warm-up stage and performed transient heat transfer analysis and thermal stress analysis. Zhang et al. [15] investigated thick film resistors (TFRs, R8241, Heraeus) on 430 stainless steel (SS) substrates for strain sensor applications. Mirjavadi et al. [16,17] studied the nonlinear vibration of nanobeams with metal cores and piezoelectric sheets and discussed the vibration behavior of nanobeams under forced vibration. Fiori et al. [18] took advantage of an energy-based failure criterion to numerically analyze the reliability of the bond–pad interconnection. Mazzei et al. [19] studied the failure mechanism of pad peeling through a wire shear test and a finite element model. Lau et al. [20] used a two-dimensional plane strain unit to perform a nonlinear transient simulation of the gold wire bonding on the Cu/low-K and performed a reliability analysis combining the bonding process and the pure wire pull test. Che et al. [21] constructed a material parameterized model and analyzed the failure mode of pull force. Chen et al. [22] discussed and compared the copper and the gold wire bonding processes on the high-power LEDs by numerical simulation. The results have disclosed that higher stress/strain in the bond pad and the ohmic contact layer is induced during the copper wire bonding process.

In this paper, the results of reliability tests conforming to the AEC-Q100 standard are analyzed with the typical pressure sensor of a new energy vehicle as an example. This sensor has high requirements for reliability as well as operating environment [23], such as humidity, dust, water, EMC, and harmful gas erosion [24,25,26]. In addition, as for many automotive devices, the requirements of vibration and shock will be much higher than those of consumer electronic products. In this paper, wire pull and shear test models based on finite-element analysis (FEA) are established to evaluate the reliability by investigating the failure mode and mechanism of gold wire bonding, and the effectiveness of the model is verified by experiments. The results show that the weak link that is most prone to fatigue failure is the gold wire. Therefore, it is worthwhile to appropriately design parameters to prevent the gold wire from fatiguing. We will describe the article in the following sections:

Section 1: The test scheme of shear test and wire pull test.

Section 2: Establishing simulation model of shear test and wire pull test.

Section 3: Research and analysis of experimental results.

Section 4: Research conclusions of the article.

## 2. Experimental Procedure

### 2.1. When Wire Pull Test

The wire pull test, an important method to detect the bonding strength of the gold wire, can be used to recognize the stress and strain state to ascertain the weak position. According to the MIL-STD-883 method 2011 standard, the wire pull test is performed in a uniaxial push–pull tester with an extremely small force and displacement scale. An upward pull force is applied under the bond alloy wire to pull the bond away from the chip surface, and the force is measured. As shown in Figure 1, pull force is gradually applied vertically through the pull hook with a loading speed of 500 μm/s at a distance of 550 μm from the ball bond until failure. A total of 70 samples were divided into 7 groups to test at the same testing condition for statistical analysis.

### 2.2. Wire Bond Shear

The strength of gold wire bonding can be measured by the wire bond shear test, which is an indicator of bonding quality and is of great significance to sensor reliability. The shear force at failure is measured according to the AEC-Q100-001 test standard [27]. The equipment includes: a test sample loading device, a shear arm with a shear head at the end of the shear arm, and a reading device that can measure the shear force. Initially, the shear head should be located on the side of the golden ball to be cut. The shear head should be pushed horizontally towards the gold ball of the tape to be cut until the gold ball is pushed away from the bond pad. Under the same experimental conditions, destructive experiments are performed on 4 sensors. The shear test is carried out as shown in Figure 2, and the shear tool is loaded with a speed of 200 μm/s.

## 3. Model and Simulation

Those tests can be applied to measure the pull force at failure; however, it fails to catch the detailed stress evolution during the pulling. The stress distribution and numerical values can be obtained by the FEM to study the possible failure mode and mechanism.

### 3.1. Simulation Model and Material Properties

The commercial software ABAQUS is used for FEA. A simplified model is used for calculation accuracy, and the parameters are consistent with the data parameters of the specific structure, with details shown in Figure 3. The simplified model structure mainly includes bonding wires (gold wires), substrate, chip, positive and negative electrodes, plastic casing, and packaging silica gel. The structures and dimensions are shown in Figure 4 and Table 1, respectively. In addition, the commercial software Hypermesh is used for meshing. The minimum element width, thickness, and length are 0.7 mm, 0.9 mm, and 3 mm, respectively, in the substrate. The model has 46,890 elements, and the material properties are shown in Table 2. Ceramic substrates and chips adopt elastic behavior, silica gel adopts viscoelastic behavior shown in Table 3, gold wire adopts elastic-plastic behavior, and solder adopts viscoplasticity behavior.

### 3.2. Boundary Conditions

#### 3.2.1. Wire Pull Test

The model shown in Figure 4 is used to simulate the wire pull test, and the following parameters are determined concerning the sensor: 3.5 mm wire loop length and 30 μm diameter. Appropriate boundary conditions are set at the wedge bond, and the bottom surface of the ball bond is constrained by all degrees of freedom (DOF). To simulate the behavior of the hook, an annular rigid body is set in the model with a vertical upward velocity of 500 μm/s.

#### 3.2.2. Wire Bond Shear

The model shown in Figure 4 is used to simulate the wire bond shear test, and the shear strain and the process of detaching from the pad were analyzed. Appropriate boundary conditions (B.C.) are defined at the ball bond and the wedge locations. At the ball bond location, the bottom surface of the chip is constrained in all DOF. At the wedge bond, the bottom surface is constrained in all DOF. The process is explored through the bond–slip method (BSM). The stress distribution on the interface is studied according to the interfacial stress, and the tension and compression regions are confirmed.

#### 3.2.3. Mechanical Shock

Given that the sensor will be shocked in the process of application, the magnitude of the shock changes greatly. A mechanical shock test is used to determine the weak link and examine the integrity of the product structure. The definition of simulated acceleration and initial velocity refers to the JEDEC JESD22-B104 test specification. On the bottom surface of the sensor, all DOF are constrained.

#### 3.2.4. Random Vibration

When the sensor is placed in a relatively flexible device, fatigue failure will occur under random vibration conditions, such as the gold wire in the sensor. Therefore, after the sensor is designed, the fatigue lifetime of the gold wire under the random vibration test needs to be estimated. The simulated acceleration and frequency values refer to the JEDEC JESD22-B103 test specification. When vibrated in the x direction, all DOF are constrained on the y–z plane, away from the gold wire, to determine the maximum failure stress. The boundary conditions are also applied when vibrating in the y and z directions by parity of reasoning.

## 4. Results and Discussion

### 4.1. Wire Pull Test

For measurement of the pull force at failure, 150 samples are divided into 15 groups to test at the same condition. Figure 5 illustrates the distribution of force at failure is fluctuating, and the difference is about 5 g. Table 4 show that the average of different groups’ data is stable, and relative deviation is less than 6.5 g with a reasonable data dispersion. Therefore, the data selected in this work are reasonable and accurate.

Figure 6 shows the undeformed and the deformed gold wire models. The maximum pull displacement is 834 μm, and the motion state of the maximum point is similar to the hook. Figure 7a shows the stress evolution process at the ball bond. When the gold wire breaks, the stress is concentrated at the joint between the ball bond and the gold wire, and the stress is 292 Mpa. Figure 7b shows that the experimental failure location verifies this. The displacement–force curve obtained by the simulation is compared with that of the experiment, as shown in Figure 8. In the initial stage of wire pull, the gold wire has not undergone plastic deformation and is only stretched within the elastic range. The gold wire began to yield after the displacement reached 0.3 mm and then increased linearly to the maximum. The gold wire breaks when the displacement reaches 1.23 mm. The pull force at failure obtained by the simulation is 18.9 g and by the experiment is 20.5 g, in which the error is about 7.8%.

In order to study the influence of the pull location on the stretching effect, the distances of 9 μm and 18 μm by a pull on both sides of the hook are investigated at the same pulling speed. Figure 9a shows the location–displacement curve, where it can be seen that the tension displacement gradually decreases as the pull location approaches the ball bond, and it can be seen that the decreasing trend is relatively fast, and the failed force also augments. Figure 9b can also indicate that the initial slope of curve is high when approaching the wedge bond. This is mainly because the process in the initial stage of stretching is to straighten the wire, and the further distance from the ball bond leads to a larger force to straighten the wire. After the gold wire is straightened, it enters the extension stage, when it is stressed again. At this time, the gold wire undergoes plastic deformation, and the pull force linearly increases until the gold wire is broken. As shown in Figure 9b, the model verifies that the pull location has a great influence, which can explain the phenomenon of unconcentrated force at failure of the 150 samples in Figure 5. This may be caused by the inability to accurately fix the location of the hook.

For the FPP001 chip, its wire diameter is 1.18 MIL, and the minimum pull force standard is 1.0 g, as defined in the MIL-STD-883 method 2011 test specification. The sensor can pass the wire bond pull test when the pull force at failure obtained by simulation and experiment far exceeds the value.

### 4.2. Wire Bond Shear

Shear force at failure was determined through averaging the experimental results of four samples with the same test condition, and the force is 42 g, as shown in Table 5.

The process of the gold ball detaching from the interface is not a single material shear failure or interface cracking, but a mixture of material damage and interface detachment. This process through the BSM can be divided into three states: the deformation accumulation stage, the cracking stage, and the disengagement stage. As shown in Figure 10a, the maximum stress at the contact position is 204.8 Mpa, lower than the ultimate strength (220 MPa), thereby the gold ball is not destroyed. Figure 10b shows the stress distribution on the bottom surface at different moments under the deformation accumulation stage, where the stressed side is pulled and the opposite side is compressed. In addition, the stress gradually extends to the center until the deformation state ends, in which the maximum stress is 97.2 MPa. When the deformation accumulation state ends, the shear force is larger on the stressed side, where cracks occur. As shown in Figure 10c, the cracking area develops inward to the center of the bottom surface when the stress is 157.4 MPa. When the cracking developed near the center, the interface gradually began to separate. As shown in Figure 10d, the interface is entirely separated until the displacement reaches the size of the bottom surface diameter. At this time, the shear process is considered to be completed, and the maximum stress is 186.3 MPa.

The displacement–force curve of the shear process is shown in Figure 11a, where the simulation maximum shear force is 38 g. Compared with the experiment, the maximum error is 5% and the minimum error is 0. Figure 11b shows the experimental failure mode and interface morphology.

For the FPP001 chip, its ball bond diameter is 3.15 MIL, and the minimum shear strength requirement of 33.2 g needs to be satisfied, as defined in the AEC-Q100-001 test specifications. This sensor can pass the wire bond shear test when the shear force at failure obtained by simulation or test is higher than the value.

In order to further analyze the stress evolution mechanism under the three states, the influence of shear location on the interface separation process was studied. Figure 12 shows the variation trend in the centerline stress in three states at a different shear location. In the deformation accumulation state, the stress distribution on the tension side fluctuates greatly when the shear location is close to the interface, which indicates that the shear tool has a significant impact on the interface, but the stress fluctuation on the compression side is small, and the stress is concentrated on 28 MPa. The stress reduction gradient at the center of the interface is small, which indicates that the stress changes slowly. In the cracking and disengagement states, the stress fluctuates greatly, and the amplitude of the stress reaches 70 MPa. The stress showed a downward trend from the tension side to the compression side, and the trend gradually slowed down. From Figure 12d, it can be seen that when the ball bond diameter exceeds 0.04 mm, the displacement in the y-direction will decrease.

From Figure 13, it can be seen that the gradient of the displacement–force curve is steep when the shear location is far from the interface, and the shear force is also large. Furthermore, the shear force difference becomes larger as the shear displacement increases. Studies have shown that if the shear location is too close to the interface, the stress fluctuates greatly, and it is impossible to accurately study the trend of interface separation.

### 4.3. Mechanical Shock

The maximum stress point appears at the upper end of the second ball bond, where the stress is 78.1 MPa, as shown in Figure 14, and there is no plastic deformation. Figure 15a shows the fitted strain–stress curve and acceleration–strain curve. The decrease in stress in Figure 15a and the distortion in the second half of the curve in Figure 15b are mainly due to the buffering effect from the internal structure on the gold wire at the end of the shock, resulting in fluctuations in stress and strain.

### 4.4. Random Vibration

Figure 16a shows the stress contour of the gold wire when randomly vibrated in the three directions of x, y, and z. It can be seen that the gold wire is easily damaged when it is vibrated randomly in the y-direction, where the maximum stress is 1.786 Mpa. It also conforms to the asymmetric and easily deformable structure in the *y*-axis direction. Figure 16b shows the frequency–strain curve during random vibration in the y-direction. When the vibration frequency reaches 65 Hz, the strain and the stress are all the largest.

### 4.5. Degeling Test

In order to test the integrity of the interface bonding, a degeling test was performed on the bonded gold wire. The interface was visually inspected after degeling, as shown in Figure 17. The bond wire on the outpad touched the pad border, but there were no visible mechanical damages.

The damage on the edge of the substrate reaches the metal line of the chip as shown in Figure 18. It was found that the crack obviously extended to the top metal line after a detailed inspection of the red area, which would definitely reach the area below the metal line.

## 5. Conclusions

The shear failure mechanism of the wire bonding interface is analyzed by the FEM. It is verified that the shearing process is not single-material damage but an interface failure process involving complex stress changes. At the same time, the interface shear is divided into three states of deformation accumulation, cracking, and disengagement, and the evolution mechanism of interface tension and compression in three states is analyzed. It is found that the stress in the deformation accumulation state decreases from the stressed side and the compression side toward the center, and the stress in the cracking and disengagement state gradually decreases from the stressed side until it is completely separated. When the interface is separating, the failed stress is 186 MPa, and the failed shear force is 42 g. In addition, the influence of the shear location on the interface failure is also studied. The results indicate the gradient of the displacement–force curve is small, and the shear force under the same displacement is also small when far from the interface. As the shear displacement increases, the shear force difference becomes larger.

In the pull test, at the initial stage of stretching, the gold wire did not undergo plastic deformation and only stretched within the elastic range. After the displacement reaches 0.3 mm, the gold wire begins to yield and then grows linearly to reach the maximum value. When the pull displacement reaches 1.23 mm, the gold wire breaks. The failed pull force obtained by the simulation is 18.9 g, and that obtained by the test is 20.22 g. The maximum error between the simulation results and the test is 7.8%. In addition, the influence of the pull location on the stretching effect is also analyzed. The closer the pull location is to the center of the horizontal gold wire, the better the stretching effect. It is proved that the pull force is close to the center, the initial slope of the displacement–force curve is high, and the pull force is large, which is consistent with the experimental failed pull force distribution.

In addition, the reliability of the sensor under mechanical shock, random vibration, and degeling experiments is analyzed. The simulation of mechanical shock and random vibration proves that the maximum stress of the gold wire is 78.1 MPa and 1.786 MPa, respectively, and the simulation results are far below the limit of damage of the gold wire. In the degeling experiment, cracks were observed and extended to the inside of the gold wire, which would affect the reliability of the sensor.

## Figures and Tables

**Figure 1 micromachines-14-01695-f001:**
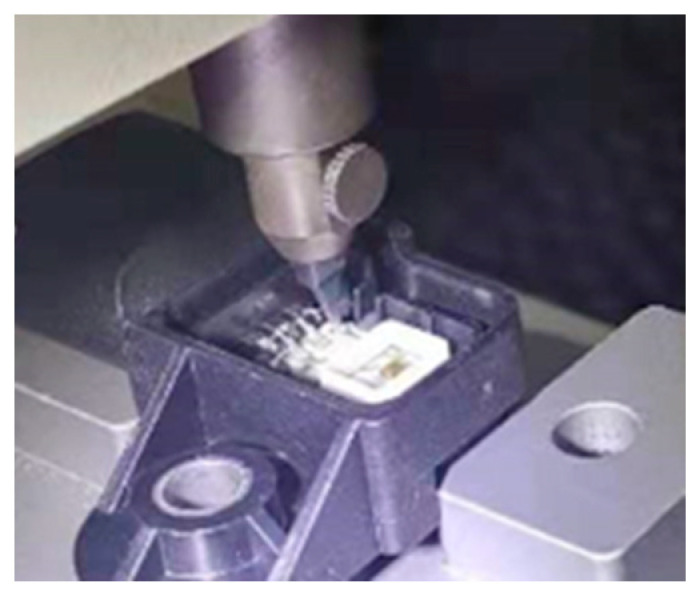
Wire pull test.

**Figure 2 micromachines-14-01695-f002:**
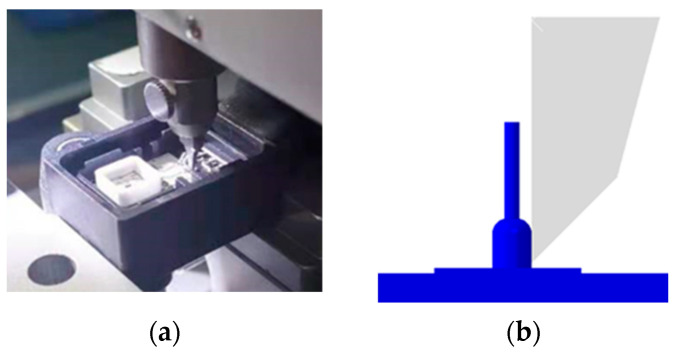
(**a**)Schematic diagram of shear test (**b**) Shear test simulation diagram.

**Figure 3 micromachines-14-01695-f003:**
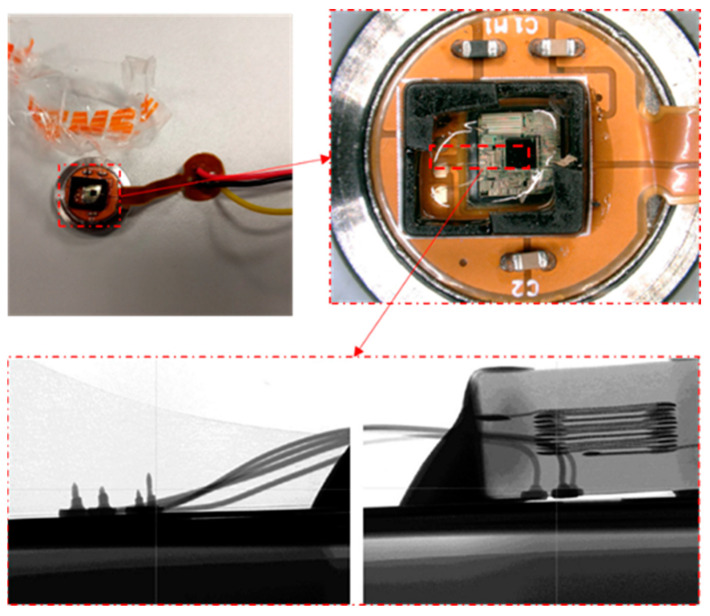
MLX90807 automotive sensor.

**Figure 4 micromachines-14-01695-f004:**
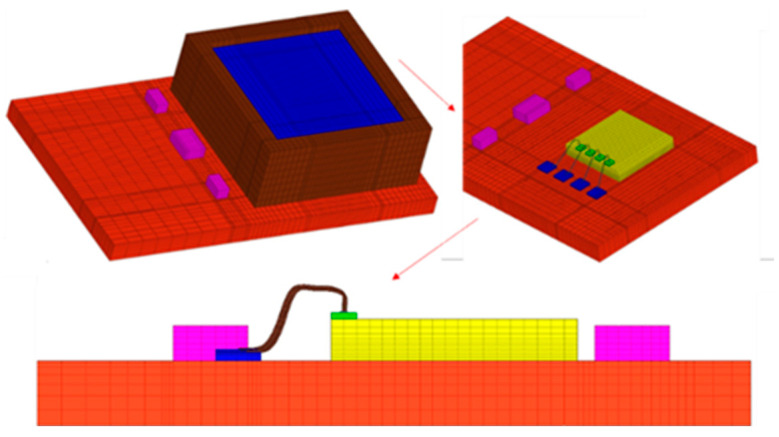
Simplified simulation model of MLX90807 automotive sensor.

**Figure 5 micromachines-14-01695-f005:**
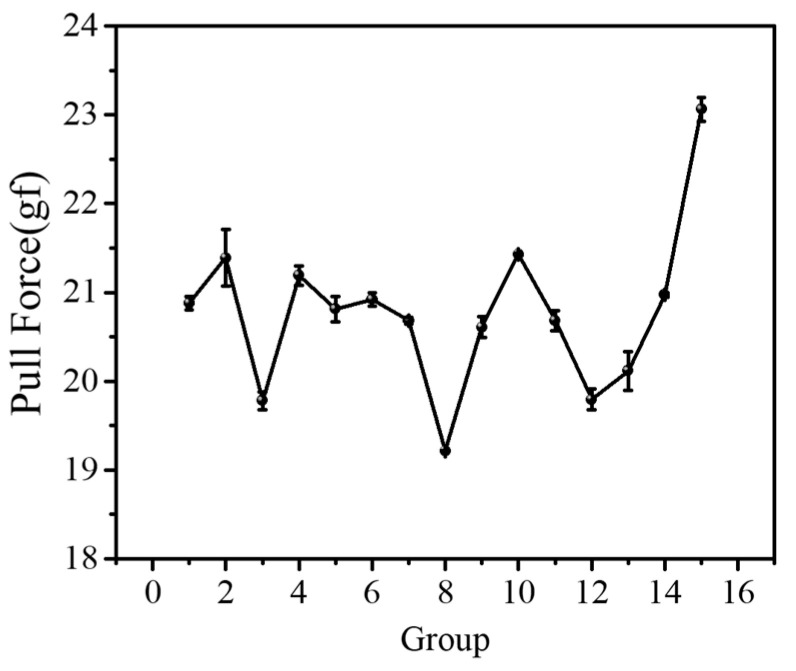
Distribution of pull force at failure.

**Figure 6 micromachines-14-01695-f006:**
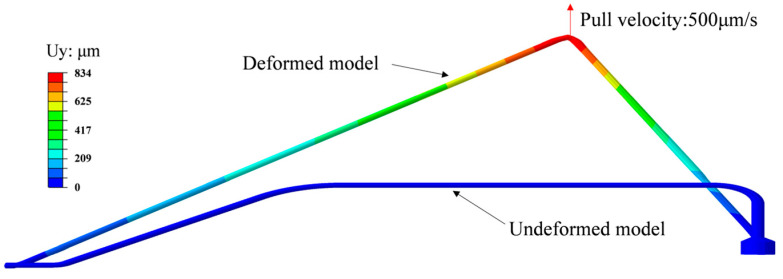
Deformed and undeformed gold wires under a speed of 500 μm/s.

**Figure 7 micromachines-14-01695-f007:**
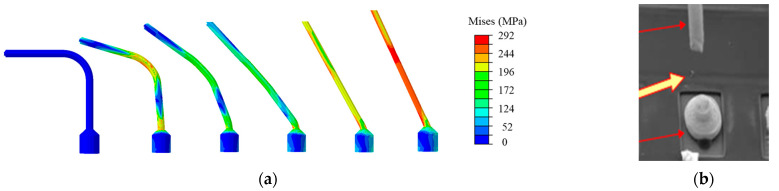
(**a**) Stress evolution process at ball bond. (**b**) The experimental failure location of pull test.

**Figure 8 micromachines-14-01695-f008:**
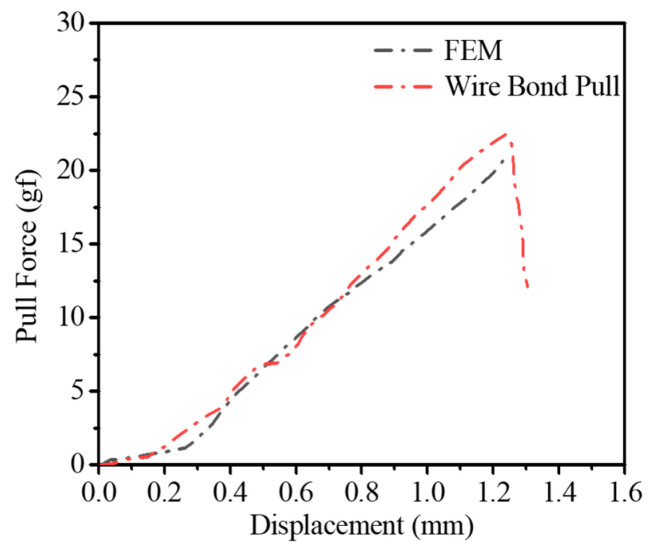
Displacement–force curve under pull test and FEM.

**Figure 9 micromachines-14-01695-f009:**
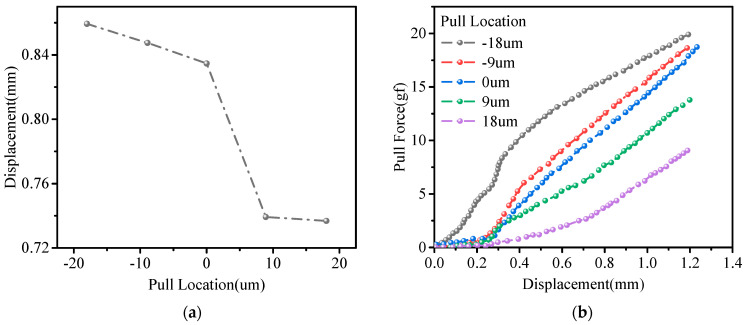
(**a**) Displacement-pull location curve. (**b**) The relationship between displacement and pull at failure at different locations.

**Figure 10 micromachines-14-01695-f010:**
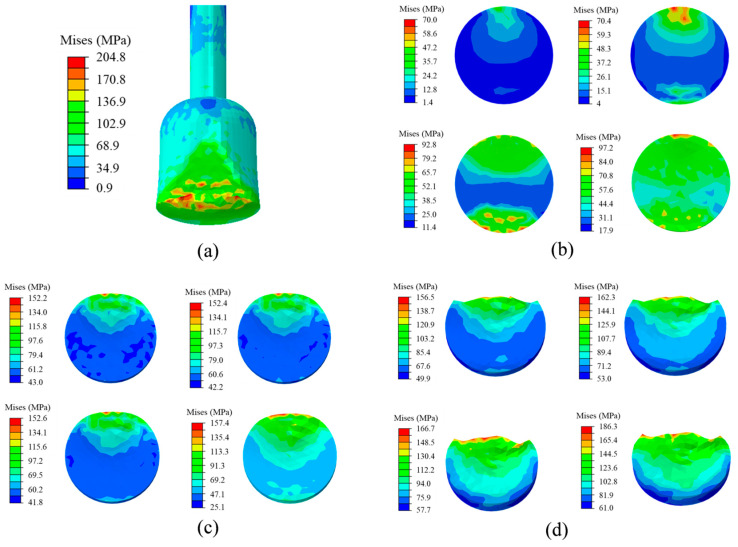
Interfacial stress distribution under different conditions. (**a**) Contact surface; (**b**) deformation accumulation; (**c**) cracking; (**d**) disengagement.

**Figure 11 micromachines-14-01695-f011:**
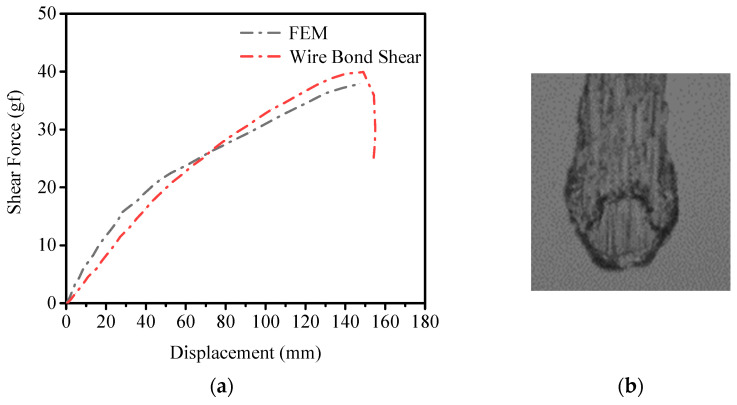
(**a**) Displacement–shear force curve under shear test and finite element simulation. (**b**) The experimental failure location of shear test.

**Figure 12 micromachines-14-01695-f012:**
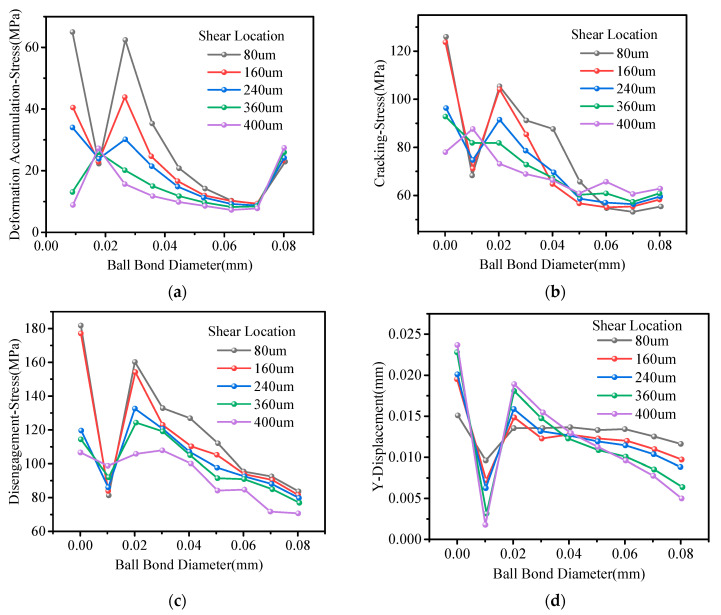
The stress evolution curve of the interface centerline under the three states. (**a**) Deformation accumulation; (**b**) cracking; (**c**) disengagement; (**d**) y-direction displacement.

**Figure 13 micromachines-14-01695-f013:**
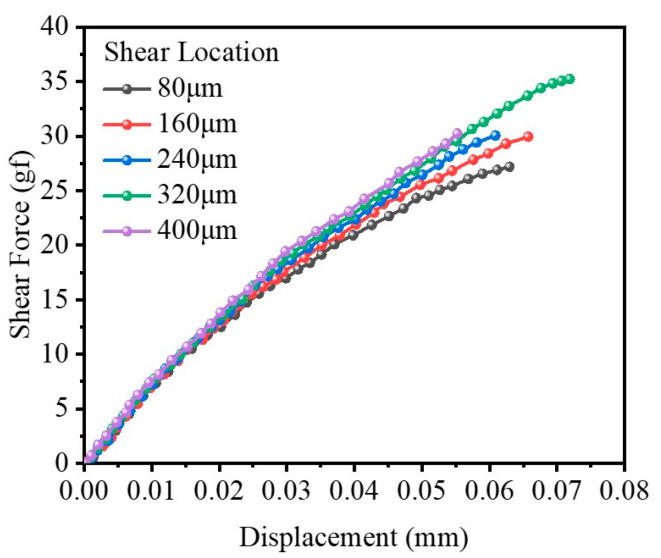
Displacement and shear force curves at different shear locations.

**Figure 14 micromachines-14-01695-f014:**
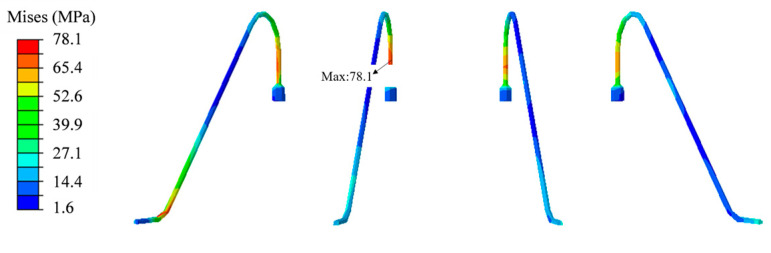
Stress contour of gold wire under mechanical shock.

**Figure 15 micromachines-14-01695-f015:**
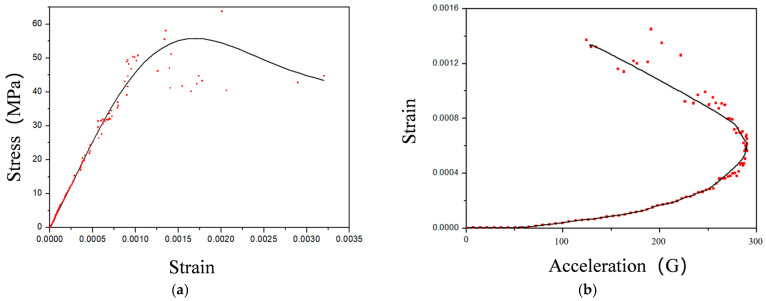
(**a**) Stress–strain curve. (**b**) Gold wire acceleration–strain curve under mechanical shock.

**Figure 16 micromachines-14-01695-f016:**
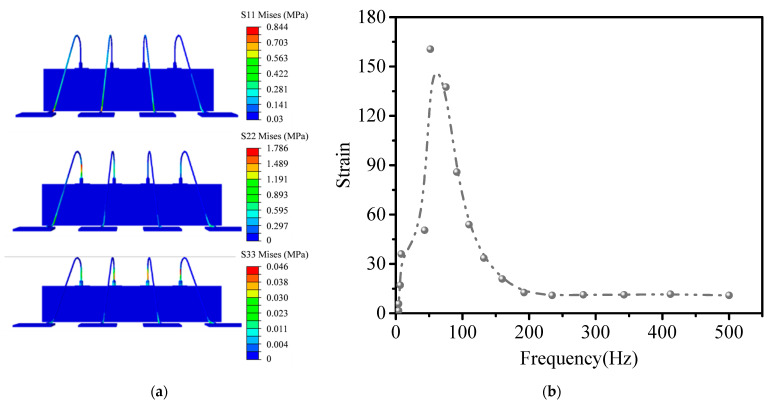
(**a**) The stress under random vibration in the three directions of x, y, and z. (**b**) Frequency–strain curve.

**Figure 17 micromachines-14-01695-f017:**
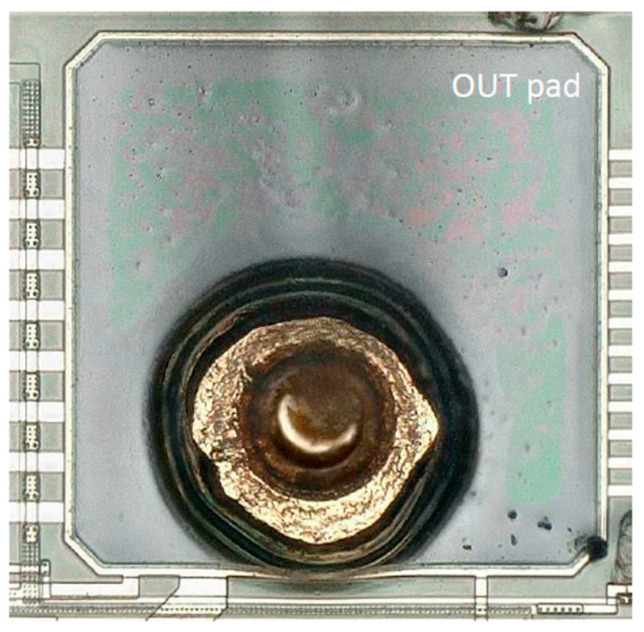
The interface after gold wire bonding degeling.

**Figure 18 micromachines-14-01695-f018:**
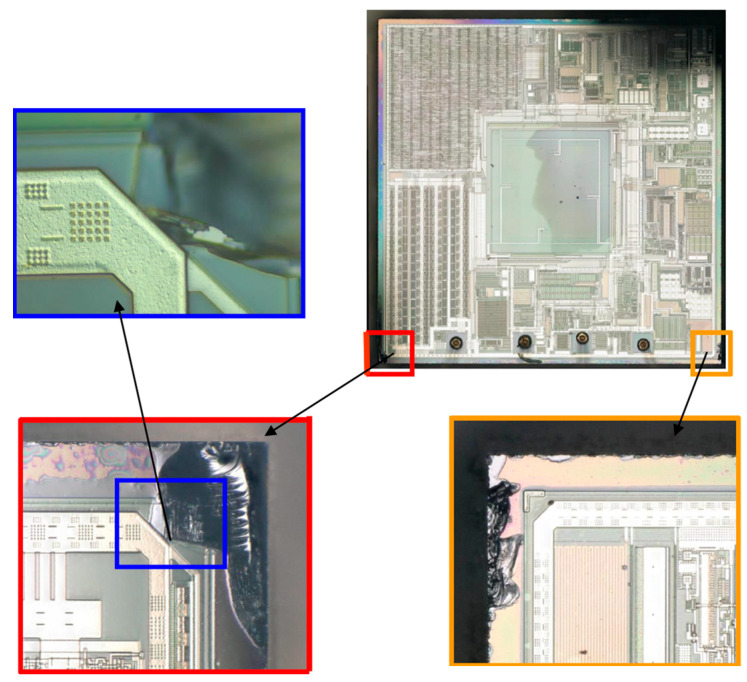
Crack initiation in the sensor.

**Table 1 micromachines-14-01695-t001:** Automotive sensor size parameters.

Automotive Sensor Size Parameters	Structure (Unit: mm^3^)
Basal plate	180 × 110 × 10
Chip	38 × 38 × 6.5
Chip solder Solder	4 × 3 × 1 7 × 7 × 1
Wire diameter	0.03
Silica gel	74 × 74 × 31

**Table 2 micromachines-14-01695-t002:** Material parameters of sensor components [28,29,30,31].

Materials	Young’s Modulus (MPa)	Poisson’s Ratio	Yield Stress (MPa)	CTE (ppm/°C)
Substrate	300,000	0.26	/	5.8
Chip	190,000,000	0.28	/	2.4
Gold wire	79,000	0.42	270	14.1
Silica gel	/	0.495	/	30
Solder	56,000	0.38	34.2	22
Cover	13,100	0.3	/	2.6

CTE: coefficient of thermal expansion.

**Table 3 micromachines-14-01695-t003:** Viscoelastic parameters of silica gel.

	Times	Modulus (MPa)
1	0	26.979
2	0.542	19.042
3	1.652	1.618
4	2.886	1.649
5	2.939	1.649
6	2.945	1.653
7	42.96	0.75
8	INF	0.518

**Table 4 micromachines-14-01695-t004:** Average and variance of 15 groups.

Group	Average (g)	Variance
1	20.7563	4.032
2	21.163	6.33
3	19.9845	1.342
4	21.344	3.628
5	20.417	3.208
6	20.9918	3.458
7	20.377	3.872
8	19.5038	4.184
9	20.4275	3.78
10	21.7535	2.726
11	20.924	2.612
12	20.216	5.884
13	20.4825	4.248
14	20.763	3.846
15	22.0763	5.262

**Table 5 micromachines-14-01695-t005:** Failed shear force.

Serial Number	1	2	3	4	5	6	7	8	9	10
Failed shear (grams)	42.6	43.8	41.8	39.8	40.1	41.3	40.8	42.4	42.1	43.2

## Data Availability

The data that support the findings of this study are available from the corresponding author upon reasonable request.

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
