# Peer review of "Failure Analysis for Gold Wire Bonding of Sensor Packaging Based on Experimental and Numerical Methods"

_micromachines, 2023, doi:10.3390/mi14091695_

Round 1
Reviewer 1 Report
my comments are annotated in the PDF

Author Response
Response to Reviewer 1 Comments
Dear editor and reviewer,
We highly appreciate your helpful suggestions and comments on our manuscript entitled “Failure Analysis and Reliability Evaluation for Gold Wire Bonding of Automotive Sensor Packaging Based on Experimental and Numerical Methods” (micromachines-2504336). The constructive suggestions are truly valuable for further improving our manuscript. We have revised our manuscript following your suggestions carefully. The highlighted revisions are described in the following.
Reviewer #1:
- Tittle is too long, please shorten.
Response:Thanks for your suggestion. We have revised our Tittle section following your suggestions carefully. The highlighted revisions are described in the following.
Tittle: Failure Analysis for Gold Wire Bonding of Sensor Packaging Based on Experimental and Numerical Methods
- The speed is influencing the failure mode, did you vary the speed in the shear testing?
Response:Many thanks. The purpose of shear testing is to check the rationality of bonding parameters, so it is necessary to observe the interface topography with the same shear force to judge whether the bonding meets the standard. Therefore, we increase the results of shear tests with different bonding parameters
|
Bonding |
Bonding Force (g) |
||||
|
640 |
660 |
680 |
700 |
720 |
|
|
60 |
CK1 |
||||
|
65 |
CK2 |
||||
|
75 |
CK3 |
||||
|
85 |
CK4 |
||||
|
90 |
CK5 |
||||
- why are you using the word simplified?.
Response:Many thanks. The sensor also contains many small structures, which cannot be completely reproduced by simulation model, so only a simplified model containing key components is established
- How did you model the shape of the wire?
Response:Many thanks. The bonding wire model is based on actual working conditions. After bonding the first solder joint, the bonding machine first straightened the bonding wire and then moved horizontally to the second solder joint for bonding.
- unit is mm3 ?
Response:Thanks for your suggestion. We have revised the table following your suggestions carefully. The highlighted revisions are described in the following.
|
Automotive Sensor Size Parameters |
Structure (Unit: mm3) |
|
Basal plate |
180 x110 x10 |
|
Chip |
38 x 38 x 6.5 |
|
Chip solder Solder |
4 x3 x1 7 x7 x1 |
|
Wire diameter |
0.03 |
|
Silica gel |
74 x 74 x 31 |
- where did you get these nr's? add references.
Response:Thanks for your suggestion. We will add the following reference files. The highlighted revisions are described in the following.
[1]M. Ikonen, “Power Cycling Lifetime Estimation of IGBT Power Modules Based on Chip Temperature Modeling.” Ph.D. Thesis, Lappeenranta University of Technology, Lappeenranta, Finland, 2012.
[2] C. Durand, M. Klingler, D.Coutellier, H. Naceur, “Power cycling reliability of power module: A survey.” IEEE Trans. Device Mater. Reliab. 2016, 16, 80–97.
[3] J. Bielen, J.J. Gommans, F. Theunis, “Prediction of high cycle fatigue in aluminum bond wires: A physics of failure approach combining experiments and multi-physics simulations.” In Proceedings of the 7th IEEE International Conference on Thermal, Mechanical and Multiphysics Simulation and Experiments in Micro-Electronics and Micro-Systems, Como, Italy, 24–26 April 2006; pp. 1–7.
[4] S. Ramminger, N. Seliger, G.Wachutka, “Reliability model for Al wire bonds subjected to heel crack failures.” Microelectron. Reliab. 2000, 40, 1521–1525.
- This cannot be true
Response:Thanks for your suggestion. We have revised the data following your suggestions carefully. The highlighted revisions are described in the following.
|
Materials |
Young’s Modulus (MPa) |
Poisson's Ratio |
Yield Stress (MPa) |
CTE (ppm/°C) |
|
Substrate |
300000 |
0.26 |
/ |
5.8 |
|
Chip |
190000000 |
0.28 |
/ |
2.4 |
|
Gold wire |
79000 |
0.42 |
270 |
14.1 |
|
Silica gel |
/ |
0.495 |
/ |
30 |
|
Solder |
56000 |
0.38 |
34.2 |
22 |
|
Cover |
13100 |
0.3 |
/ |
2.6 |
- add picture of test results: where does it break?
Response:Thanks for your suggestion. We will add the following failure photos.
- Why is group 15 so much higher?
Response:Many thanks. This may be due to fluctuations in the power capacity of the bonding machine during package bonding, but the greater the shear force, the higher the interface strength naturally
- this cannot be seen in the figure?
Response:Many thanks. This column of values represents the variance. This data is not included in the figure, but is directly displayed in the table after calculation by the variance formula.
- nice results!
Response:Many thanks.
- ???
Response:FPP001 represents the Product Number of the chip.
- please add pictures of the failures
Response:Thanks for your suggestion. We will add the following failure photos.
- 4 samples is not much, should be 10!
Response:Many thanks. We have increased the samples to 10 following your suggestions carefully. The highlighted revisions are described in the following.
|
Serial Number |
1 |
2 |
3 |
4 |
5 |
6 |
7 |
8 |
9 |
10 |
|
Failed shear (grams) |
42.6 |
43.8 |
41.8 |
39.8 |
40.1 |
41.3 |
40.8 |
42.4 |
42.1 |
43.2 |
- Something should be wrong at the horizontal axis. G-level is huge.
Response:Thanks for your suggestion. The acceleration was incorrectly recorded and should have been between 0 and 300G, We have revised this figure 15b following your suggestions carefully.
- why did you add this part?
Response:Many thanks. In addition to the study of bonding, some reliability problems, including random vibration, mechanical shock and Degeling Test, are also mentioned in this paper. If you think it is a bit cumbersome, we can delete this part of the content.
- it is kind of okeish in nr of reference, but there is many more available. Please extend the literature review.
Response:Thanks for your suggestion. We will increase the number of references.

Reviewer 2 Report
1. For the following mentioned content: the influence of the pull location and shear location,These two points were not mentioned in the abstract.
2. Keyword : ‘wire shear and pull test’ should be’ wire pull and shear test’
3. ‘If the bonding force is set improperly, it will cause circuit breakage or even chip damage. Circuit breakage and even chip damage often occur if the bonding force is set improperly.’The content of these two sentences is repeated in the article.
4. ‘the results of reliability tests conforming to the AEC-Q standard’,as can be seen from the previous text, it should be AEC-Q100 standard.
5. ‘The results show that the weak link that is most prone to fatigue failure is the gold wire. Therefore, it is worthwhile to appropriately design parameters to prevent the gold wire.’From these two sentences, it can be seen that ‘design parameters to prevent fatigue failure of the gold wire.’
6. In the last paragraph of the introduction, a brief list of the following sections should be provided.
7. ‘Those tests can be applied to measure the pull force at failure, however, it fails to catch the detailed stress evolution during the pulling.’The overview of the two experimental tests mentioned earlier is not comprehensive enough.
8. Figures 3 and 4 can indicate the relevant structures in the figure.
9. ‘Figure 5 illustrates the distribution of force at failure is fluctuation, and the difference is almost about 9 grams.’From the graph, it can be seen that the difference cannot reach 9 grams.
10. ‘Besides, the average of different groups' data is stabilization, and relative deviation is less than 2 grams’.The relative difference between 3 and 15 in Table 4 exceeds 2 grams.
11. ‘the force is determined to be 20.5 grams shown in Table 4’,However, it cannot be observed from Figure 4.
12. The vertical axis of Figure 5 should be pull force.
13. ‘The stress is concentrated at the wire neck, and the maximum stress obtained by simulation is 351 MPa.’There is no relevant content in Figure 6.
14. ‘The pull force at failure obtained by the simulation is 18.9 grams, and by the experiment is 20.5 grams, in which the error is about 6.5%’,The error obtained from the calculation is 7.8%.
15. The vertical axis of Figure 8 should be pull force.
16. ‘Figure 9a shows the location-displacement curve, where it can be seen that the tension displacement gradually decreases as the pull location approaches the ball bond’,From Figure 9a, it can be seen that the decreasing trend is relatively fast.
17. The description of some data in Figure 10 in the text is not very accurate.
18. ‘Figure 12d shows that when the shear location is farther from the interface, the displacement in the y-direction is greater’ From Figure 12d, it can be seen that when Ball Bond Diameter exceeds 0.04mm,the displacement in the y-direction will decrease.
19. ‘Figure 13 shows the displacement-shear force curve at different shear locations. The result indicates the gradient of the displacement-force curve is small when the shear location is far from the interface, and the shear force is also little’ From Figure 13, it can be seen that the gradient of the displacement-force curve is big when the shear location is far from the interface, and the shear force is also big.
20. ‘The maximum stress point appears at the upper end of the second ball bond, where the stress is 77 MPa, as shown in Figure 14’From Figure 14, it can be seen that the stress is 78.1 MPa.
21. ‘Figure 16 shows the stress contour of the gold wire when randomly vibrated in the three directions of x, y, and z.’ it should be Figure 16a.
22. ‘It can be seen that the gold wire is easily damaged when it is vibrated randomly in the y-direction, where the maximum stress is 1.08 MPa’ it should be 1.786 MPa.
23. The title of Figure 16a is incomplete.
24. There are also many erroneous data in the conclusion.
Author Response
Response to Reviewer 2 Comments
Dear editor and reviewer,
We highly appreciate your helpful suggestions and comments on our manuscript entitled “Failure Analysis and Reliability Evaluation for Gold Wire Bonding of Automotive Sensor Packaging Based on Experimental and Numerical Methods” (micromachines-2504336). The constructive suggestions are truly valuable for further improving our manuscript. We have revised our manuscript following your suggestions carefully. The highlighted revisions are described in the following.
Reviewer #2:
- For the following mentioned content: the influence of the pull location and shear location,These two points were not mentioned in the abstract.
Response:Thanks for your suggestion. We will revise our Abstract section following your suggestions carefully. The highlighted revisions are described in the following.
Abstract: There is an increasing demand for automotive sensors, whereas complex working en-vironments may easily lead to failure. Wire pull and shear test models based on finite-element analysis are established to evaluate the reliability by investigating the failure mode and mecha-nism of gold wire bonding. The effect of shear force position and pull force position on failure is also analyzed. The bonding failure was verified by experiments, which is con-sistent with the simulation result. The results show that: 1) The three-dimensional quantitative modeling reveals the process of bonding delamination and stress concentration. 2) The bond-ing-slip method (BSM) is adopted in the gold ball detaching process. The concept of three states including deformation accumulation, cracking and disengagement was put forward to reveal the interface stress evolution trend according to the shear testing results. Besides, the results indicate that in the interface, the stress in the deformation accumulation state decreases from the tensile side (or compression side) to the center, and the stress in the cracking and disengagement state reduces gradually from the tensile side to edge. when the interface is completely separated, the failed shear force concentrates on 42 grams. The concept and theory proposed in this work can effectively reveal the failure mechanism of bonding interface and help to establish a new failure criterion.
- Keyword : ‘wire shear and pull test’ should be’ wire pull and shear test’?
Response:Many thanks. We will revise our Keyword following your suggestions carefully. The highlighted revisions are described in the following.
Keywords: failure mechanism and reliability; wire bonding; wire pull and shear test
- ‘If the bonding force is set improperly, it will cause circuit breakage or even chip damage. Circuit breakage and even chip damage often occur if the bonding force is set improperly.’The content of these two sentences is repeated in the article.
Response:Thanks for pointing out this. We will remove the same description in the line 64-67. The highlighted revisions are removed in the following.
Line 64-67: Wire bonding technology, as the most basic interconnection technology, has a great influ-ence on device reliability. If the bonding force is set improperly, it will cause circuit break-age or even chip damage. Circuit breakage and even chip damage often occur if the bond-ing force is set improperly.
- ‘the results of reliability tests conforming to the AEC-Q standard’,as can be seen from the previous text, it should be AEC-Q100 standard.
Response:Many thanks. We did carry out the experiment according to AEC-Q100 standard. We will revise this following your suggestions carefully.
Line 93: the results of reliability tests conforming to the AEC-Q100 standard.
- ‘The results show that the weak link that is most prone to fatigue failure is the gold wire. Therefore, it is worthwhile to appropriately design parameters to prevent the gold wire.’From these two sentences, it can be seen that ‘design parameters to prevent fatigue failure of the gold wire.’
Response:Yes. The reliability of the bonding line is very important to the sensor, and reasonable design parameters can improve the reliability
- In the last paragraph of the introduction, a brief list of the following sections should be provided.
Response:Thanks for your suggestion. We will provide a brief list of the following sections. The highlighted revisions are described in the following.
We will describe the article from the following sections:
Section Ⅰ: The test scheme of shear test and wire pull test.
Section Ⅱ: Establishing simulation model of shear test and wire pull test.
Section Ⅲ: Research and analysis of experimental results
Section Ⅳ: Research conclusions of the article
- ‘Those tests can be applied to measure the pull force at failure, however, it fails to catch the detailed stress evolution during the pulling.’The overview of the two experimental tests mentioned earlier is not comprehensive enough.
Response:Many thanks. We will add descriptions of two experimental tests following your suggestions carefully. The highlighted revisions are described in the following.
Line106-113: The wire pull test, an important method to detect the bonding strength of the gold wire, can be used to recognize the stress and strain state to ascertain the weak position. According to the MIL-STD-883 Method 2011 standard, the wire pull test is performed in a uniaxial push-pull tester with an extremely small force and displacement scale. An upward pull force is applied under the bond alloy wire to pull the bond away from the chip surface, and the force is measured. As shown in Figure 1, pull force is gradually applied vertically through the pull hook with a loading speed of 500 μm/s at a distance of 550 um from the ball bond until failure. 70 samples were divided into 7 groups to test at the same testing condition for statistic analysis.
Line117-122: The strength of gold wire bonding can be measured by the wire bond shear test, which is an indicator of bonding quality and is of great significance to sensor reliability. The shear force at failure is measured according to the AEC-Q100-001 test standard. The equipment includes: a test sample loading device, a shear arm with a shear head at the end of the shear arm, and a reading device that can measure the shear force. Initially, the shear head should be located on the side of the golden ball to be cut. The shear head should be pushed horizontally towards the gold ball of the tape to be cut until the gold ball is pushed away from the bond pad. Under the same experimental conditions, destructive experiments are performed on 4 sensors. The shear test is carried out as shown in Figure 2, and the shear tool is loaded with a speed of 200 μm/s.
- Figures 3 and 4 can indicate the relevant structures in the figure.
Response:Many thanks.
- ‘Figure 5 illustrates the distribution of force at failure is fluctuation, and the difference is almost about 9 grams.’From the graph, it can be seen that the difference cannot reach 9 grams.
Response:Thanks for your suggestion. We carefully recalculate and review the relevant data. We will revise this conclusion following your suggestions carefully. The highlighted revisions are described in the following.
Line183: Figure 5 illustrates the distribution of force at failure is fluctuation, and the difference is almost about 5 grams.
- ‘Besides, the average of different groups' data is stabilization, and relative deviation is less than 2 grams’.The relative difference between 3 and 15 in Table 4 exceeds 2 grams.
Response:Thanks for your suggestion. We carefully recalculate and review the relevant data. We will revise this conclusion following your suggestions carefully. The highlighted revisions are described in the following.
Line184: Besides, the average of different groups' data is stabilization, and relative deviation is less than 6.5 grams.
- ‘the force is determined to be 20.5 grams shown in Table 4’,However, it cannot be observed from Figure 4.
Response:Thanks for your suggestion. We have carefully recalculated and reviewed the relevant data and believe that this conclusion is hasty. We will remove this conclusion following your suggestions carefully. The highlighted revisions are described in the following.
Line185: the force is determined to be 20.5 grams shown in Table 4.
- The vertical axis of Figure 5 should be pull force.
Response:Thanks for your suggestion. We will revise this figure following your suggestions carefully.
- ‘The stress is concentrated at the wire neck, and the maximum stress obtained by simulation is 351 MPa.’There is no relevant content in Figure 6.
Response:Thanks for your suggestion. We carefully recalculated and reviewed the relevant data and concluded that this data could not be obtained from the simulation results. We will remove this conclusion following your suggestions carefully. The highlighted revisions are described in the following.
Line192: The stress is concentrated at the wire neck, and the maximum stress obtained by simulation is 351 MPa.
- ‘The pull force at failure obtained by the simulation is 18.9 grams, and by the experiment is 20.5 grams, in which the error is about 6.5%’,The error obtained from the calculation is 7.8%.
Response:Thanks for your suggestion. We carefully recalculate and review the relevant data. We will revise this conclusion following your suggestions carefully. The highlighted revisions are described in the following.
Line201: The pull force at failure obtained by the simulation is 18.9 grams, and by the experiment is 20.5 grams, in which the error is about 7.8%.
- The vertical axis of Figure 8 should be pull force.
Response: Thanks for your suggestion. We will revise this figure following your suggestions carefully.
- ‘Figure 9a shows the location-displacement curve, where it can be seen that the tension displacement gradually decreases as the pull location approaches the ball bond’,From Figure 9a, it can be seen that the decreasing trend is relatively fast.
Response:Thanks for your suggestion. We will revise this conclusion following your suggestions carefully. The highlighted revisions are described in the following.
Line208-210: Figure 9a shows the location-displacement curve, where it can be seen that the tension displacement gradually decreases as the pull location approaches the ball bond, and it can be seen that the decreasing trend is relatively fast.
- The description of some data in Figure 10 in the text is not very accurate.
Response:Thanks for your suggestion. We carefully recalculate and review the relevant data. We will revise some data following your suggestions carefully. The highlighted revisions are described in the following.
Line234: As shown in Figure 10a, the maximum stress at the contact position is 204.8 Mpa.
Line239: In addition, the stress gradually extends to the center until the deformation state ends, in which the maximum stress is 97.2 MPa.
Line241: As shown in Figure 10c, the cracking area develops inward to the center of the bottom surface, when the stress is 157.4 MPa.
Line245: At this time, the shear process is considered to be completed, and the maximum stress is 186.3 MPa.
- ‘Figure 12d shows that when the shear location is farther from the interface, the displacement in the y-direction is greater’ From Figure 12d, it can be seen that when Ball Bond Diameter exceeds 0.04mm,the displacement in the y-direction will decrease.
Response:Thanks for your suggestion. We will revise this conclusion following your suggestions carefully. The highlighted revisions are described in the following.
Line268: From Figure 12d, it can be seen that when Ball Bond Diameter exceeds 0.04mm,the displacement in the y-direction will decrease.
- ‘Figure 13 shows the displacement-shear force curve at different shear locations. The result indicates the gradient of the displacement-force curve is small when the shear location is far from the interface, and the shear force is also little’ From Figure 13, it can be seen that the gradient of the displacement-force curve is big when the shear location is far from the interface, and the shear force is also big.
Response:Thanks for your suggestion. We will revise this conclusion following your suggestions carefully. The highlighted revisions are described in the following.
Line274: From Figure 13, it can be seen that the gradient of the displacement-force curve is big when the shear location is far from the interface, and the shear force is also big.
- ‘The maximum stress point appears at the upper end of the second ball bond, where the stress is 77 MPa, as shown in Figure 14’From Figure 14, it can be seen that the stress is 78.1 MPa.
Response:Thanks for your suggestion. We carefully recalculate and review the relevant data. We will revise this conclusion following your suggestions carefully. The highlighted revisions are described in the following.
Line284: The maximum stress point appears at the upper end of the second ball bond, where the stress is 78.1 MPa, as shown in Figure 14.
- ‘Figure 16 shows the stress contour of the gold wire when randomly vibrated in the three directions of x, y, and z.’ it should be Figure 16a.
Response:Thanks for your suggestion. We will revise this conclusion following your suggestions carefully. The highlighted revisions are described in the following.
Line292: Figure 16a shows the stress contour of the gold wire when randomly vibrated in the three directions of x, y, and z.
- ‘It can be seen that the gold wire is easily damaged when it is vibrated randomly in the y-direction, where the maximum stress is 1.08 MPa’ it should be 1.786 MPa.
Response:Thanks for your suggestion. We carefully recalculate and review the relevant data. We will revise this conclusion following your suggestions carefully. The highlighted revisions are described in the following.
Line294: It can be seen that the gold wire is easily damaged when it is vibrated randomly in the y-direction, where the maximum stress is 1.786 Mpa.
- The title of Figure 16a is incomplete.
Response:Thanks for your suggestion. We will revise the title of figure 16a following your suggestions carefully. The highlighted revisions are described in the following.
- There are also many erroneous data in the conclusion.
Response:Thanks for your suggestion. We carefully recalculate and review the relevant data. We will revise this conclusion following your suggestions carefully. The highlighted revisions are described in the following.
Line332: The maximum error between the simulation results and the test is 7.8%.
Line339: The simulation of mechanical shock and random vibration proves that the maximum stress of the gold wire is respectively 78.1 MPa and1.786 MPa,

Round 2
Reviewer 2 Report
I have no other comments. Congratulations!